# Antibacterial Activity and Mechanism of Linalool against *Shigella sonnei* and Its Application in Lettuce

**DOI:** 10.3390/foods11203160

**Published:** 2022-10-11

**Authors:** Ruiying Su, Peng Guo, Ziruo Zhang, Jingzi Wang, Xinyi Guo, Du Guo, Yutang Wang, Xin Lü, Chao Shi

**Affiliations:** 1College of Food Science and Engineering, Northwest A&F University, Yangling 712100, China; 2School of Science, Xi’an Jiaotong-Liverpool University, Suzhou 215123, China

**Keywords:** linalool, *Shigella sonnei*, antibacterial activity, reactive oxygen species, lettuce, sensory evaluation

## Abstract

*Shigella sonnei (S. sonnei*) infection accounted for approximately 75% of annual outbreaks of shigellosis, with the vast majority of outbreaks due to the consumption of contaminated foods (e.g., fresh vegetables, potato salad, fish, beef, etc.). Thus, we investigated the antibacterial effect and mechanism of linalool on *S. sonnei* and evaluated the effect of linalool on the sensory quality of lettuce. The minimum inhibitory concentration (MIC) of linalool against *S. sonnei* ATCC 25931 was 1.5 mg/mL. *S. sonnei* was treated with linalool at 1× MIC for 30 min and the amount of bacteria was decreased below the detection limit (1 CFU/mL) in phosphate-buffered saline (PBS) and Luria-Bertani (LB) medium. The bacterial content of the lettuce surface was reduced by 4.33 log CFU/cm^2^ after soaking with linalool at 2× MIC. Treatment with linalool led to increased intracellular reactive oxygen species (ROS) levels, decreased intracellular adenosine-triphosphate (ATP) content, increased membrane lipid oxidation, damaged cell membrane integrity, and hyperpolarized cell membrane potential in *S. sonnei*. The application of linalool to lettuce had no effect on the color of lettuce compared to the control. The sensory evaluation results showed that linalool had an acceptable effect on the sensory quality of lettuce. These findings indicate that linalool played an antibacterial effect against *S. sonnei* and had potential as a natural antimicrobial for the inhibition of this foodborne pathogen.

## 1. Introduction

*Shigella* is a kind of non-spore, non-pod, non-flagellate Gram-negative bacteria widely found in fruits, vegetables, milk, and other foods [1]. *Shigella* infection can cause intestinal infectious disease bacterial malaria [2]. Bacterial malaria is usually transmitted by water and food, so it is widespread and transmitted rapidly. Moreover, it is easy to spread among people and seriously affects human health [3]. For the past few years, *Shigella sonnei* (*S. sonnei*) showed an obvious increase trend and became the dominant bacteria of *Shigella* spp. [4]. There have been several outbreaks of disease due to *S. sonnei* infection. In 2011, 293 patients with *S. sonnei* were isolated in designated hospitals in Shanghai [5].

At present, the sterilization effect of ultraviolet ray, heat treatment, ozone, and chemical disinfectant on *S. sonnei* has been reported [6,7,8]. Although heat treatment has a better sterilization effect, it has a certain influence on the sensory quality and nutritional value of food. Heat treatment can lead to amino acid degradation or a Maillard reaction with sugars, resulting in a decrease in the amount of free amino acids and a decrease in the nutritional value of milk [9,10]. Overexposure to commonly used disinfectants may lead to tolerance or resistance in *S. sonnei* to these compounds, and can lead to residues in food, with the consumers’ low acceptance level. It has been reported that free chlorine and chlorine dioxide disinfectant have high residuals after use [11,12]. Some new sterilization equipment such as pulse sterilization, radiation sterilization, and other methods of high cost are still in the stage of theoretical research [13,14].

Recently, with the improvement in people’s living standards, people are increasingly pursuing natural, green, and less impact on the food sensory and nutritional quality sterilization methods. The use of natural compounds to control foodborne pathogens in food and food production has been studied [15]. Linalool (C_10_H_18_O) is a monoterpene alcohol widely found in monocotyledonous and dicotyledonous plants, which is recognized as a safe ingredient by generally recognized as safe (GRAS) [16,17]. Linalool has a variety of biological activities such as anti-anxiety, anti-convulsion, anti-cancer, analgesic, anti-oxidation, and hypolipidemic [18]. Herman et al. found that linalool had antibacterial activity against *Staphylococcus aureus* [19]. However, the effect and mechanism of linalool on *S. sonnei* have not been studied.

The study investigated the antibacterial effect and antibacterial mechanism of linalool against *S. sonnei* and the inactivation effect against *S. sonnei* on contaminated lettuce. The purpose is to provide a theoretical basis for the application of linalool in lettuce washing, which may be used as a natural antimicrobial agent for the control of *S. sonnei* in vegetables and other areas of the food supply chain, thereby reducing the risk of *S. sonnei* infection.

## 2. Materials and Methods

### 2.1. Reagents

Linalool (density 0.87 g/mL at 25 °C, CAS, 78-70-86) was purchased from Sigma-Aldrich (Shanghai, China). Luria-Bertani (LB) agar and broth were acquired from Land Bridge Technology Co. (Beijing, China). All other chemicals and reagents were analytically pure.

### 2.2. Bacterial Strains and Culture Conditions

*S. sonnei* ATCC 25931 was obtained from the American Type Culture Collection (ATCC, Manassas, VA, USA). The strain was stored in LB broth with 25% glycerol (*v*/*v*) at −80 °C. A single colony of *S. sonnei* was collected by the inoculation ring and inoculated in LB broth for 14 h at 37 °C with shaking at 130 rpm.

### 2.3. Effects of Linalool on the Inhibitory Effect of S. sonnei

#### 2.3.1. Determination of MIC and Minimum Bactericidal Concentration (MBC)

The MIC of linalool against *S. sonnei* was determined by using the broth microdilution method described in the Clinical and Laboratory Standards Institute guidelines (CLSI, 2019), with some modifications. *S. sonnei* was centrifuged (8000× *g*, 5 min, 4 °C), washed, and resuspended in LB broth to achieve a bacterial suspension with an optical density at 600 nm (OD_600_) = 0.5 for a bacterial suspension concentration of 3 × 10^8^ CFU/mL. The bacterial suspension was diluted to 5 × 10^5^ CFU/mL. A total of 100 µL of bacterial solution with linalool was added to 96-well plates to give final concentrations of 0 (control), 0.25, 0.375, 0.5, 0.75, 1, 1.5, 2, and 3 mg/mL of linalool. The samples were incubated at 37 °C for 24 h and then the OD_600_ was measured. The MIC was defined as the lowest antimicrobial concentration of linalool corresponding to an OD_600_ change of <0.05. The bacterial suspension from each well that showed inhibition was plated onto LB agar cultured at 37 °C for 48 h. The minimum concentration of linalool above or equal to the MIC and capable of no colony growth was defined as the MBC.

#### 2.3.2. Growth Curves

Growth curve determination was carried out as described in Zheng et al. (2019), with minor modifications [20]. LB broth was used to prepare linalool solutions at 0 (control), 1/8× MIC, 3/16× MIC, 1/4× MIC, 3/8× MIC, 1/2× MIC, 3/4× MIC, 1× MIC, 3/2× MIC, 2× MIC, and 3× MIC. The culture medium of *S. sonnei* was diluted to ~2 × 10^6^ CFU/mL. A total of 125 μL of bacterial suspension and linalool solution were added to each well. The fully-automated Bioscreen Plate Reader (Labsystems, Helsinki, Finland) was used to measure the OD_600_ values every 1 h at 37 °C for 24 h continuously.

#### 2.3.3. Inactivation Effect of Linalool on *S. sonnei* in LB Broth and Phosphate-Buffered Sa-Line (PBS)

The suspension of *S. sonnei* (5 × 10^6^ CFU/mL) was mixed with linalool solution (0 (control), 1/2× MIC, 1× MIC and 2× MIC) and incubated at 37 °C in LB broth and PBS. *S. sonnei* in LB broth were plated onto LB agar after 0, 30, 60, 120, 240, and 480 min for bacterial count. *S. sonnei* in PBS was plated onto LB agar after 0, 10, 20, 30, 40, 60, 90, and 120 min for bacterial count. The samples were incubated at 37 °C for 24 h and counted.

### 2.4. Antibacterial Mechanism of Linalool on S. sonnei

#### 2.4.1. Intracellular ROS Levels

The intracellular ROS level of *S. sonnei* was determined by 2′,7′-dichlorodihydrofluorescein diacetate (DCFH-DA) with reference to Akhtar et al. (2021) [21]. The DCFH-DA at 5 μM was added to the bacterial solution (1 × 10^7^ CFU/mL) and incubated at 37 °C for 10 min in the dark. Linalool (at final concentrations of 0 (control), 1/2× MIC, 1× MIC, and 3/2× MIC) were added to the bacterial solution and incubated at 37 °C for 10 min and then centrifuged (10,000× *g*, 10 min, 4 °C). Fluorescence was detected at 488 nm and 525 nm with a microplate reader (Spectra Max M2; Molecular Devices, San Jose, CA, USA). The live cells were diluted with PBS and plated onto LB agar at 37 °C for 24 h. The calculation formula of intracellular ROS is as follows:$$\mathrm{Intracellular}\;\mathrm{ROS}=\frac{\mathrm{Intracellular}\;\mathrm{relative}\;\mathrm{fluorescence}\;\mathrm{intensity}}{\mathrm{Bacterial}\;\mathrm{counts}}$$

#### 2.4.2. Intracellular ATP Concentrations

The experiment was conducted according to the method of Li et al. (2014) [22]. Samples of 20 mL bacterial suspensions (3 × 10^8^ CFU/mL) containing different linalool concentrations of 3/2× MIC, 1× MIC, 1/2× MIC, and 0 (control) were cultured at 37 °C for 10 min. Then, the samples were treated with an ultrasonic sonifier (JY92-II, Ningbo Scientz Biotechnology, Ningbo, China) in an ice bath. The supernatant was collected by high-speed centrifugation (12,000× *g*, 5 min, 4 °C), immediately transferred, and stored on ice until use. We added 100 μL each of the supernatant or standard liquid added with an ATP working solution to a 96-well white plate for measuring absorbance by using the multifunctional microplate meter (Spectra Max M2; Molecular Devices, San Jose, CA, USA). The concentrations of intracellular ATP were calculated using standard curves and the measured absorbance values.

#### 2.4.3. Extracellular MDA Content

As described by Alminderej et al. (2021), the change of malondialdehyde (MDA) in the *S. sonnei* was quantified using the lipid peroxidation MDA assay kit with slight modification [23]. Linalool was mixed with the bacterial solution (3 × 10^8^ CFU/mL) until the final concentrations of linalool was 0 (control), 1/4× MIC, 1/2× MIC, and 1× MIC and incubated at 37 °C for 10 min. After centrifugation (8000× *g*, 5 min, 4 °C), the supernatant was mixed with the MDA working solution at a concentration of 0.67% (*w*/*v*) and boiled at 100 °C for 1 h. The absorbance at 450, 532, and 600 nm was measured using the multifunctional microplate detector (Spectra Max M2; Molecular Devices, San Jose, CA, USA) after the samples were cooled to room temperature. The calculation method of extracellular MDA content is as follows.
MDA (nmol/L) = (12.9 × (ΔA532 − ΔA600) − 2.58 × ΔA450) × V_1_ ÷ V_2_

V_1_: Total volume of sample and test fluid;

V_2_: The volume of the sample.

#### 2.4.4. Field-Emission Scanning Electron Microscopy (FESEM)-Based Observations

FESEM was performed as described by Guo et al. (2019), with some modifications [24]. *S. sonnei* (3 × 10^8^ CFU/mL) was treated with linalool at concentrations equivalent to 0 (control), 1/2× MIC, 1× MIC, and 3/2× MIC and incubated at 37 °C for 2 h and 4 h. Cells were then harvested by centrifugation (5000× *g*, 5 min, 4 °C) and rinsed twice with PBS. The cells were fixed overnight with 2.5% (*v*/*v*) glutaraldehyde. After centrifugation (5000× *g*, 5 min, 4 °C), glutaraldehyde was removed and fixed again for 6 h in accordance with the above method. The cells were continuously dehydrated by the water ethanol gradient (30%, 50%, 70%, 80%, 90%, 100%) for 10 min. The cells were fixed on the FESEM bracket and observed by scanning electron microscopy (S-4800; Hitachi, Tokyo, Japan).

#### 2.4.5. Membrane Potential

The experiment was carried out according to Liu et al. (2020) [25]. The bacterial suspension (3 × 10^8^ CFU/mL) was added to the black plate and incubated for 30 min. The bacterial suspension was added to 1 μM bis-(1,3-dibutyl barbituric acid) trimethine oxonol (DiBAC4(3); Molecular Probes, Portland, OR, USA) as a membrane potential sensitive fluorescent probe and incubated at 37 °C for 30 min. Linalool was added into the samples so that the concentrations of linalool were 0 (control), 1/2× MIC, 1× MIC, and 2/3× MIC. After 5 min, the multifunctional microplate detector (Spectra Max M2; Molecular Devices, San Jose, CA, USA) was used to detect the fluorescence intensity. The excitation wavelength was 492 nm, and the emission wavelength was 515 nm.

### 2.5. Effects of Linalool on Lettuce Leaves

#### 2.5.1. Inactivation Effect of Linalool on *S. sonnei* on Leaf Surface of Lettuce

The experimental method of Sadekuzzaman et al. (2017) was used to test the inactivation effect of linalool on *S. sonnei* on the surface of lettuce, but slightly modified [26]. Lettuces were bought from a local market (Yangling, China). Lettuces were cut into squares with a size of 5.0 cm × 5.0 cm, soaked in 75% alcohol and cleaned for 30 min, then dried. According to method 2.2, the bacterial suspension was prepared and inoculated on the surface of lettuce with the inoculation amount of 1 × 10^7^ CFU/cm^2^. The samples were treated with 1/2× MIC, 1× MIC, 3/2× MIC, and 2× MIC linalool for 5, 15, and 30 min at 25 °C and transferred to a 12 × 11 cm sterile homogeneous bag for homogenous beating for 2 min (10 times per second, 25 °C). The samples were diluted and plated onto LB agar, and incubated at 37 °C for 24 h for counting.

#### 2.5.2. Superficial Color

Lettuces were cut into squares with a size of 5.0 cm × 5.0 cm, then washed with water and dried. Lettuce leaves were soaked in different concentrations of linalool solution (0 (control), 1/2× MIC, 1× MIC, 3/2× MIC, and 2× MIC) for 30 min and then removed. The color of the leaves was measured using a colorimeter (CS-820, Color Spectrum Technology, Hangzhou, China). Ten random areas of the lettuce portions were measured for each treatment. The instrument recorded the CIELAB color components L*, a* and b*, and ∆E was obtained by calculation.

#### 2.5.3. Organoleptic Quality: Overall Visual Quality (OVQ)

Overall visual quality (OVQ) was based on the methodology of Miceli et al. (2019) [27]. A sensory panel of 20 trained people was formed by judges, members of the food engineering group, and members with experience in sensory evaluation of leafy vegetables. A sensory trained panel assessed the color, texture (the soft crispness of lettuce), odor, and physiological disorders (browning of veins and leaf edges of lettuce leaves) of the linalool-treated (0 (control), 1/2× MIC, 1× MIC, 2/3× MIC, and 2× MIC) lettuce leaves. The sensory standards were as follows: color, where 5 = very good/bright white, 3 = moderate/duskiness, and 1 = poor/dim; texture, where 5 = very good/fresh/crispy, 3 = moderate crispness, and 1 = poor/limp; odor, where 5 = very good/fresh/full characteristic, 3 = moderate, and 1 = poor/none/not typical; physiological disorders, where 5 = none, 3 = moderate, and 1 = severe; and overall visual quality (OVQ), where 5 = very good/fresh appearance, 3 = moderate, and 1 = poor/no fresh appearance. Each lettuce was presented in random order, one at a time, to the judges who conducted the independent evaluation. 

### 2.6. Statistical Analysis

All experiments were performed in triplicate. Data were presented as the mean ± standard deviation (SD) (*n* = 3). Statistical analyses were carried out using SPSS software (version 26.0; IBM Corp., Armonk, NY, USA). One-way ANOVA was used for the analysis of variance, and Tukey and Fisher’s least significant difference test (LSD) were used to analyze the differences between the experimental and control groups, with *p* < 0.05 and *p* < 0.01 as statistically significant differences.

## 3. Results

### 3.1. Effects of Linalool on the Inhibitory Effect of S. sonnei

#### 3.1.1. MIC and MBC

Linalool had an inhibitory effect on *S. sonnei*. The minimum inhibitory concentration (MIC) and MBC of linalool against *S. sonnei* were 1.5 mg/mL.

#### 3.1.2. Growth Curve

The results of linalool on the *S. sonnei* growth curve determination are shown in Figure 1. *S. sonnei* completely inhibited the growth at linalool concentrations of 3× MIC, 2× MIC, 3/2× MIC, and 1× MIC. Exposed to the 3/4× MIC of linalool, *S. sonnei* lagged to 7 h. After 1/2× MIC to 3/16× MIC linalool treatments, the maximum growth values of *S. sonnei* decreased compared to the control. When the concentration of linalool was reduced to 1/8× MIC, the growth curve of *S. sonnei* was consistent with that of the control.

#### 3.1.3. Antibacterial Curve Assay

Inactivation of *S. sonnei* by linalool in the LB broth and PBS is shown in Figure 2. The total number of living cells in all treatment groups was initially 6.0 log CFU/mL. When the *S. sonnei* was cultured in LB broth (Figure 2A), the cells without linalool (control) increased by 2.3 log CFU/mL at 480 min. *S. sonnei* ATCC 25931 was treated with linalool at 1× MIC and 2× MIC for 30 min and the amount of bacteria was reduced to below the detection limit (1 CFU/mL). When *S. sonnei* was cultured in PBS (Figure 2B), the number of living cells in both the control group and 1/2 × MIC linalool treatment group remained stable at 6 log CFU/mL within 120 min. After treatment with linalool at 1× MIC and 2× MIC, the number of *S. sonnei* decreased to an undetectable level (1 CFU/mL) after 10 and 20 min, respectively.

### 3.2. Antibacterial Mechanism of Linalool on S. sonnei

#### 3.2.1. Intracellular ROS Level

Figure 3 shows that the effect of linalool on the intracellular ROS level of *S. sonnei*. The cells without linalool (control) on the intracellular ROS level were 6.67 ± 3.42. After treatment with linalool at 1/2× MIC, the intracellular ROS level increased to 9.00 ± 3.10. When the *S. sonnei* was exposed to 1× MIC and 3/2× MIC linalool, the intracellular ROS level of bacteria was significantly (*p* < 0.01) increased.

#### 3.2.2. Intracellular ATP Content

We established a linear relationship between the relative luminescence units and intracellular ATP concentration (y = 68011x + 2070, R^2^ = 0.9994, the standard curve is not shown). The intracellular ATP concentration of the untreated group (control) was 5.65 ± 0.48 μM. After linalool at 1/2× MIC, 1× MIC, and 3/2× MIC treatment, the intracellular ATP concentrations decreased to 1.21 ± 0.13, 0.34 ± 0.02, and 0.31 ± 0.01 μM, respectively. *S. sonnei* treated with linalool of 1/2× MIC, 1× MIC, and 3/2× MIC showed significant reductions (*p* < 0.01) of intracellular ATP concentration compared with the control (Figure 4).

#### 3.2.3. Extracellular Malondialdehyde (MDA) Content

As shown in Figure 5, cells treated by linalool significantly increased the MDA concentration (*p* < 0.01). While after treatment with linalool at 1/4× MIC, 1/2× MIC, and 1× MIC, the MDA concentrations increased to 0.22 ± 0.01, 0.33 ± 0.01, and 0.96 ± 0.01 nmol/mL, respectively. There was a positive correlation between the concentration of MDA in *S. sonnei* and the amount of linalool used.

#### 3.2.4. FESEM Observations

The results in Figure 6 show that linalool treatment changed the cell morphology of *S. sonnei*. Without linalool treatment, *S. sonnei* was rod-shaped with a smooth and full surface shape (Figure 6A,E). *S. sonnei* treated with 1/2× MIC linalool showed surface depressions and cell morphology shrinkage after 2 h and 4 h (Figure 6B,F). The 1× MIC linalool treatment resulted in cell surface contractions intensifying and beginning to rupture (Figure 6C,G). Cell rupture increased with increasing linalool concentration (Figure 6D,H).

#### 3.2.5. Membrane Potential

The effect of linalool on the *S. sonnei* membrane potential is shown in Figure 7. Linalool caused hyperpolarization of the *S. sonnei* cell membrane potential (the fluorescence values were negative). The relative fluorescence intensity of the untreated group was 0 and remained stable within 20 min. *S. sonnei* treated with linalool at 3/2× MIC showed a sustained increase in cell membrane potential hyperpolarization within 20 min.

### 3.3. Effects of Linalool on Lettuce Leaves

#### 3.3.1. Inactivation Effect of Linalool on *S. sonnei* on the Leaf Surface of Lettuce

Figure 8 shows the changes in the quantity of *S. sonnei* on the lettuce surface after treatment with different concentrations of linalool. The initial values of *S. sonnei* cells in each treatment group were the same (~7 log CFU/cm^2^). After treatment with linalool at 1/2, 1, and 3/2× MIC for 30 min, *S. sonnei* on the lettuce surface was reduced to 6.62 ± 0.25, 5.54 ± 0.14, and 4.62 ± 0.06 log CFU/cm^2^, respectively. The number of *S. sonnei* on lettuce leaves exposed to linalool at 2× MIC continued to decrease within 30 min, and decreased by 1.56, 3.07 and 4.33 log CFU/cm^2^ at 5, 15 and 30 min, respectively.

#### 3.3.2. Surface Color

The effect of linalool on the chromaticity of lettuce leaves is shown in Table 1. The L* value of lettuce leaves treated with 1/2× MIC linalool was 56.86 ± 1.29, which showed no difference compared to the control group. As the concentration of linalool increased, there was still no difference in the L*, a*, b* values of the lettuce leaves. These results indicate that there was no significant effect on the ∆E values of lettuce leaves soaked in linalool solution.

#### 3.3.3. Organoleptic Quality: Overall Visual Quality (OVQ)

The OVQ value of the untreated group was 4.81 ± 0.16, and that of the 2× MIC treatment group decreased to 3.89 ± 0.20. The OVQ values of lettuce leaves treated with linalool were all decreased compared to the control group (Figure 9 and Figure 10).

## 4. Discussion

In this study, linalool at 1× MIC (1.5 mg/mL) showed antibacterial activity against *S. sonnei*. The inhibitory effect of natural active substances on *S. sonnei* has rarely been mentioned in previous studies. Bagamboula et al. (2004) reported that the 1× MIC of oregano essential oil against *S. sonnei* was 0.5 % *w*/*v* (5.00 mg/mL) [25]. The study showed that 431 μg/mL of linalool had an inhibitory effect on *Pseudomonas aeruginosa* [28]. These indicated that linalool had an effective antimicrobial activity against *S. sonnei.*

The study showed that linalool inhibited *S**. sonnei* in the LB broth and PBS media (Figure 2) and lettuce leaves (Figure 8). Kang et al. (2022) showed that citral at 0.4 mg/mL reduced *Yersinia enterocolitica* in the PBS medium to below the limit of detection within 30 min, while it took 180 min for citral at 0.4 mg/mL to reduce *Y. enterocolitica* in LB broth to below the limit of detection (<1 CFU/mL) [29]. The amount of *E. coli* treated with 2 μL/mL of camphor essential oil decreased by 3 log CFU/mL [30]. Kang et al. (2020) showed that noni extract and oregano essential oil treatment could reduce *Listeria monocytogenes* on the lettuce leaf surface by 1.84 and 1.82 log CFU/g compared with the control [31]. Kraśniewska et al. (2020) showed that the combined application of 0.025% (*v*/*v*) Spanish origanum oil and 0.1% (*v*/*v*) coriander oil reduced the amount of *Listeria monocytogenes* on the surface of vegetable filter leaves to less than 1 log CFU/mL in 24 h [32]. Viacava et al. (2018) showed that 31.6 g/L of *Thymus vulgaris* essential oil microcapsules had an antimicrobial effect on psychrotrophic bacteria on lettuce leaves, with a reduction of about 2 log CFU/g at the end of storage compared to the control [33]. The differences in the bactericidal effect of linalool in the three media may be due to differences in the growth medium and the growth state of the bacteria.

ROS is a normal product of cellular metabolism and the propagation of ROS can lead to cell damage [34]. With the increase in linalool concentration, the intracellular ROS level of bacteria increased (Figure 3). In a previous study, the intracellular ROS of *Escherichia coli* increased after 15 min of 0.625 μM monocaprin and carvacrol, resulting in cell damage or cell death [35]. Studies have also shown that the excessive production of ROS may reduce the expression of genes related to proton dynamics and accelerate the Fenton reaction. High concentrations of hydroxyl radicals, which are considered to be the main ROS species, cause cell membrane stress damage and the oxidation of biological macromolecules, leading to bacterial death [36,37]. Therefore, linalool may inhibit *S. sonnei* by increasing the intracellular ROS levels.

ATP depletion could be a potential indicator of the effect of linalool on the bacterial cell membrane permeability. In this study, linalool decreased the intracellular ATP content of *S. sonnei*, which was consistent with the results (Figure 4). The intracellular ATP concentration of *Listeria monocytogenes* decreased gradually with the increase in the treatment time by 0.8 mg/mL of limonene [38]. Similarly, Cui et al. (2015) showed that *salvia sclarea* essential oil caused a decrease in intracellular ATP in *Staphylococcus aureus* after 30 min [39]. The decrease in ATP may be due to the loss of inorganic phosphate through the damaged membrane, resulting in the change in the equilibrium of ATP hydrolysis reaction and the dissipation of proton dynamics [40]. In summary, linalool may cause increases in cell membrane permeability. MDA, as an important product of lipid oxidation, is an important indicator to verify oxidative stress to probe bacterial death. In the study, the extracellular MDA content of bacteria increased significantly with the increase in the linalool concentration (Figure 5). Ju et al. (2020) showed that the synergistic use of eugenol and citral increased the MDA content of *Aspergillus niger* [41]. Lee et al. (2017) showed that resveratrol at 20 μg/mL for 4 h resulted in significant MDA production in *Salmonella*, which was suggested to be due to oxidative damage induced by lipid peroxidation, leading to bacterial death [42]. The results of this study demonstrated the induction of oxidative stress and lipid peroxidation by linalool in *S. sonnei*, contributing to the alteration of cell membrane permeability caused by the production of MDA.

The results observed by FESEM showed that the morphology of *S. sonnei* cells had changed and collapsed after linalool treatment (Figure 6). Liu et al. (2021) showed that *E. coli* was significantly damaged after 30 min of treatment with a concentration of 0.5 μL/mL peppermint oil nanoemulsion [43]. Patra et al. (2015) found that the cell surface became rough after 12.5 mg/mL of *E. linza* L. essential oil in the treatment of *E. coli* [44]. Based on the above experimental results, the change in cell morphology may be due to a change in cell membrane permeability, resulting in the collapse of cell morphology due to the leakage of contents.

The membrane potential is involved in energy metabolism as a key factor in proton dynamics and is also associated with ATP synthesis. Liu et al. (2019) used Rhodamine fluorescence to demonstrate that linalool hyperpolarizes *Pseudomonas aeruginosa* cell membranes [45]. Luo et al. (2022) showed that the membrane potential of *Vibrio vulnificus* showed a tendency of hyperpolarization with the increase in the oregano essential oil concentration within 12 min [46]. In the study, linalool caused hyperpolarization of the cell membrane of *S. sonnei* (Figure 7). Proton dynamics of the cell may be disrupted as well as cell membrane permeability and dysfunction.

In the study, linalool had no effect on the lettuce leaf color, but there were significant differences in browning, texture, and overall visual quality (Figure 10). Similarly, studies have shown that 4% (*v*/*v*) eucalyptus oil soaked in leaves for 1 h causes yellowing of the leaves [47]. However, Gülten et al. (2010) used a sensory panel to determine the sensory quality of lettuce leaves soaked in oregano essential oil at 0.01% (*v*/*v*) for 25 min and showed no effect of oregano essential oil on the sensory quality of lettuce leaves [48]. In our study, it is possible that the addition of linalool caused a change in the pH value of the soaking solution and thus a change in the texture of the lettuce leaves was altered. It is also possible that linalool reacts with the damaged substances in fresh-cut leaves, leading to a change in the sensory quality.

The results of this study showed that linalool had an inhibitory effect on *S. sonnei*. The results of this study showed that the mechanisms of inhibition of *S. sonnei* by linalool include increases in intracellular ROS and extracellular MDA and a loss of membrane potential and ATP depletion, accompanied by the observation of gross ultrastructural damage to the cell. At the same time, linalool was able to inhibit *S. sonnei* on the surface of the lettuce, and did not deteriorate the sensory quality of the lettuce leaves. In summary, linalool has potential application in the control of contamination caused by *S. sonnei* in the food industry.

## Figures and Tables

**Figure 1 foods-11-03160-f001:**
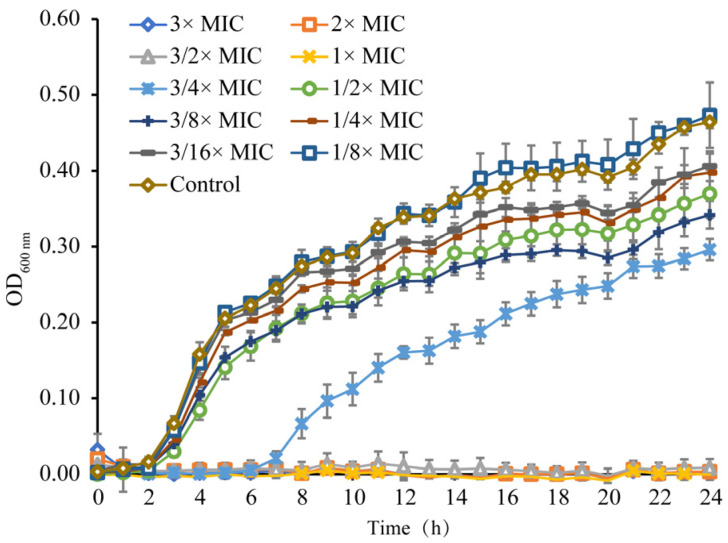
The growth curve analysis of *S. sonnei* cultured in LB broth with various concentrations of linalool. OD_600 nm_, optical density at 600 nm. Bars represent the standard deviation (*n* = 3).

**Figure 2 foods-11-03160-f002:**
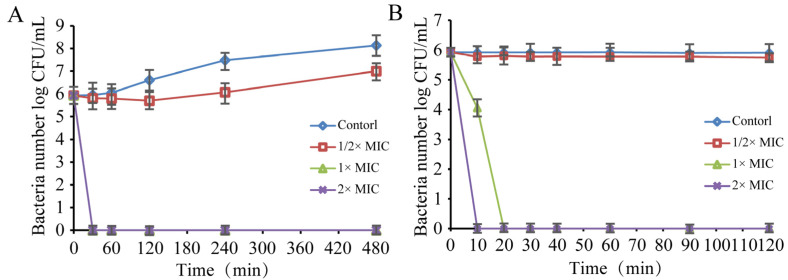
The effect of linalool on the number of *S. sonnei* in LB (**A**) and PBS (**B**) media at 37 °C. Bars represent the standard deviation (*n* = 3).

**Figure 3 foods-11-03160-f003:**
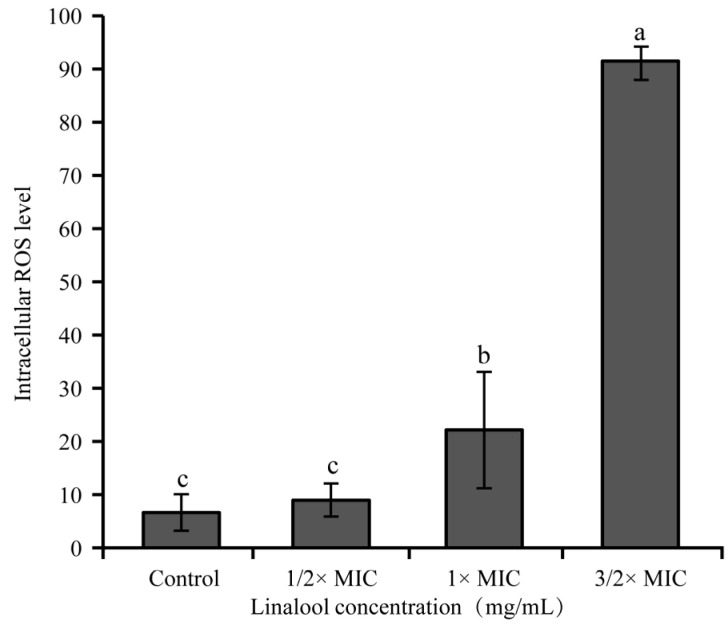
The effect of linalool on the ROS concentration in *S. sonnei*. Values represent the means of independent triplicate measurements. Bars represent the standard deviation (*n* = 3). Different letters indicate a significant difference cross the figure (*p* < 0.01).

**Figure 4 foods-11-03160-f004:**
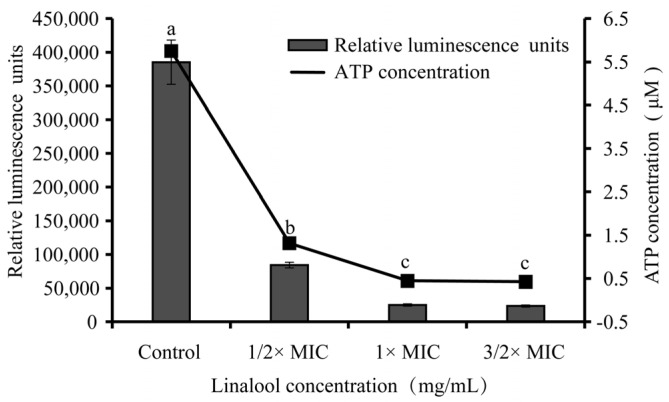
The effects of linalool on intracellular ATP production by *S. sonnei*. Values represent the means of independent triplicate measurements. Bars represent the standard deviation (*n* = 3). Different letters indicate a significant difference cross the figure (*p* < 0.01).

**Figure 5 foods-11-03160-f005:**
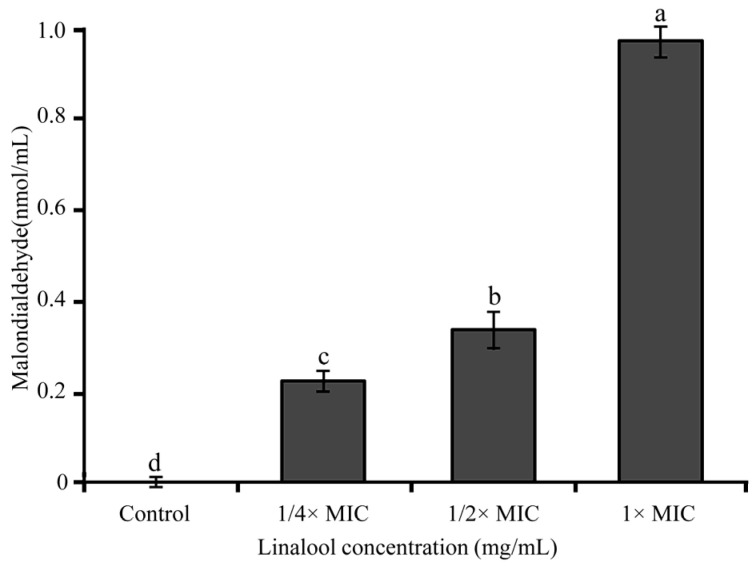
Effects of linalool on the degree of lipid oxidation of *S. sonnei*. Values represent the means of independent triplicate measurements. Bars represent the standard deviation (*n* = 3). Different letters indicate a significant difference cross the figure (*p* < 0.01).

**Figure 6 foods-11-03160-f006:**
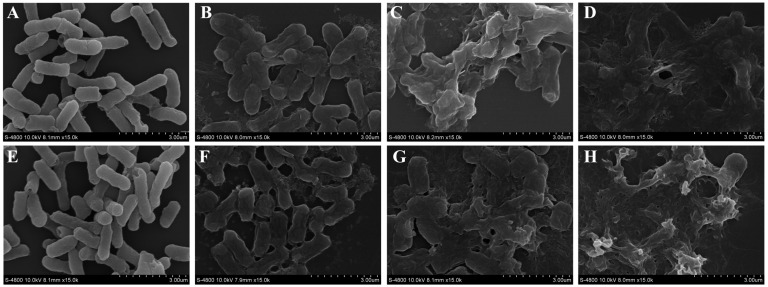
Field-emission scanning electron micrographs of *S. sonnei*. (**A**) Untreated bacterial cells at 2 h post-inoculation. (**B**) Bacterial cells treated with linalool at 1/2× MIC for 2 h. (**C**) Bacterial cells treated with linalool at 1× MIC for 2 h. (**D**) Bacterial cells treated with linalool at 3/2× MIC for 2 h. (**E**) Untreated bacterial cells at 4 h post-inoculation. (**F**) Bacterial cells treated with linalool at 1/2× MIC for 4 h. (**G**) Bacterial cells treated with linalool at 1× MIC for 4 h. (**H**) Bacterial cells treated with linalool at 3/2× MIC for 4 h.

**Figure 7 foods-11-03160-f007:**
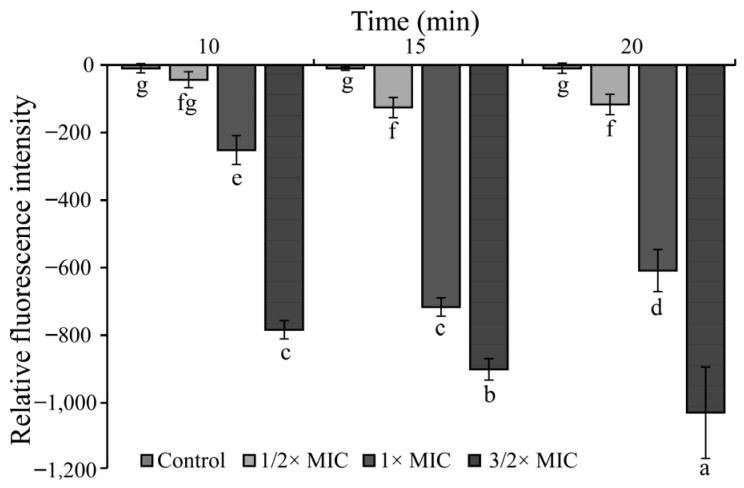
The effects of linalool on the membrane potentials of *S. sonnei*. Values represent the means of independent triplicate measurements. Bars represent the standard deviation (*n* = 3). Different letters indicate a significant difference cross the figure (*p* < 0.01).

**Figure 8 foods-11-03160-f008:**
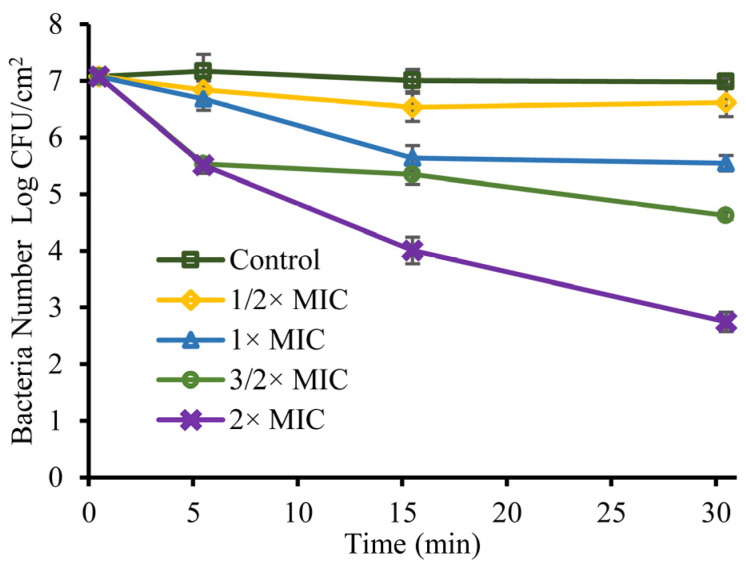
The antibacterial curve of linalool against *S. sonnei* on the lettuce surface. Bars represent the standard deviation (*n* = 3).

**Figure 9 foods-11-03160-f009:**
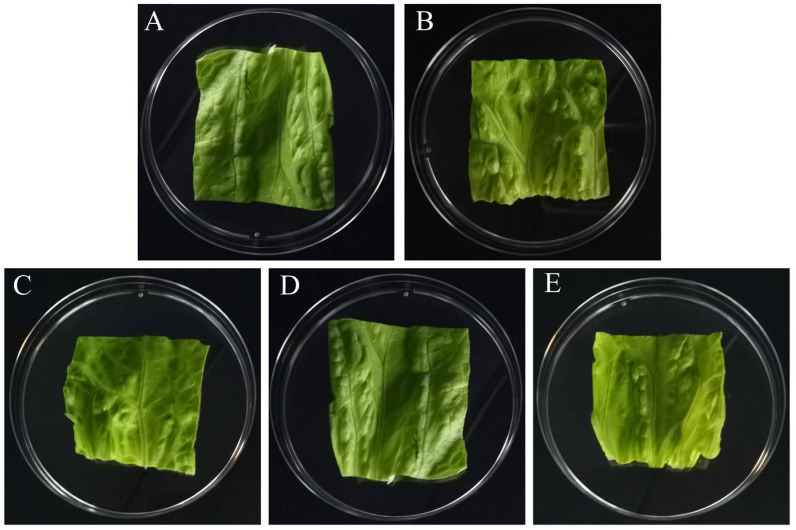
Lettuce leaves treated with different concentrations of linalool. (**A**) Untreated group. (**B**) Lettuce treated with linalool at 1/2× MIC. (**C**) Lettuce treated with linalool at 1× MIC. (**D**) Lettuce treated with linalool at 3/2× MIC. (**E**) Lettuce treated with linalool at 2× MIC.

**Figure 10 foods-11-03160-f010:**
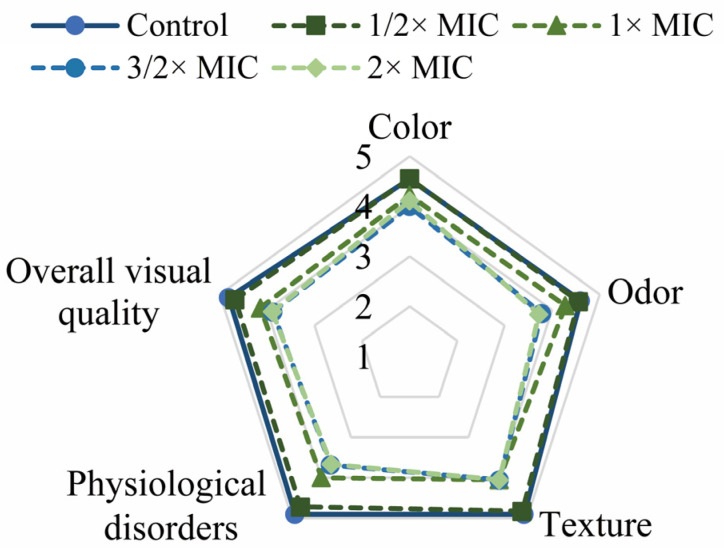
Sensory evaluation of linalool on lettuce leaves of the color, odor, texture, physiological disorders, and overall visual quality.

**Table 1 foods-11-03160-t001:** Determination of the chroma and color difference of lettuce leaves treated with different concentrations of linalool. Different letters indicate a significant difference cross the table (*p* < 0.05).

Color Parameter	Treatment				
	Control	1/2× MIC	1× MIC	3/2× MIC	2× MIC
L*	59.68 ± 1.80 ^a^	56.86 ± 1.29 ^a^	60.08 ± 0.77 ^a^	57.34 ± 3.12 ^a^	56.99 ± 3.04 ^a^
a*	−6.60 ± 0.67 ^a^	−7.65 ± 0.32 ^a^	−6.99 ± 0.19 ^a^	−6.81 ± 0.96 ^a^	−6.88 ± 1.09 ^a^
b*	19.76 ± 2.05 ^a^	22.48 ± 1.34 ^a^	22.18 ± 0.50 ^a^	20.37 ± 2.18 ^a^	21.33 ± 3.13 ^a^
ΔE	4.81 ± 0.53 ^a^	4.46 ± 0.52 ^a^	5.21 ± 0.38 ^a^	4.00 ± 0.76 ^a^	4.89 ± 0.23 ^a^

## Data Availability

The original contributions presented in the study are included in the article. Further inquiries can be directed to the corresponding author.

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
