# Peer review of "Antibacterial Activity and Mechanism of Linalool against Shigella sonnei and Its Application in Lettuce"

_foods, 2022, doi:10.3390/foods11203160_

Round 1
Reviewer 1 Report
The manuscript "Antibacterial activity and mechanism of linalool against Shigella sonnei and its application in lettuce" presents interesting research results but contains some inaccuracies which I ask the authors to clarify. The manuscript would have been clearer if in vitro studies had been presented first and then lettuce studies had been presented, now it is mixed up.
Detailed comments:
Chapter 2.3.1 - why was LB substrate used for tasks and not Mueller-Hinton, which is the standard used? Why were the MIC tests conducted for 48 hours, while others for 24 hours?
Chapter 2.3.1 - on what basis was the definition of "The MIC was defined as the lowest antimicrobial concentration of linalool corresponding to an OD 600 change of <0.05."? Why was change of 0.05 chosen, please refer to the literature.
Chapter 2.3.2 - this is a time-kill curve, not a growth curve.
line 118, 120 - samples not Samples
line 146 - the year of publication of the article is missing
Different concentrations of linalool are used in the study of mechanisms, they should be the same.
Chapter 2.5.5. - was it a trained panel? The size of the panel was small.
Since several strains of S. sonnei were not compared (which would be advisable in this type of study), it makes no sense to quote the ATCC collection number every time, just once in the methodology.
Chapter 3.1.3 - please indicate why the inactivation in the two media was compared, what is the scientific basis for the selection of these media.
There is very little discussion about the effect of natural substances such as essential oils in inhibiting the growth of bacteria on lettuce. The authors should expand on this. For example, you can add "Foods 2020, 9, 1740; doi: 10.3390 / foods9121740", "Foods 2021, 10(3), 575; https://doi.org/10.3390/foods10030575" and https://doi.org/10.1016/j.postharvbio.2018.07.004
Author Response
Plesae see the attachment.

Reviewer 2 Report
The article is well written, but it is suggested to add data on the importance of linalool application in lettuce.
Introduction: suggests adding information on the significance of applying linalool to lettuce
The methodology has to be explained in more detail. E.g.: Only 8 sensory panels were used; were they trained or untrained panels in terms of organoleptic quality? Explain the meaning of "physiological abnormalities" (line 191) and how is the texture characteristic was visually assessed. Which sensory standard was used to conduct the evaluation? Please specify.
The term "MIC" has to be defined (line 66) and the relevant concentration has to be justified.
List the statistical tests that were performed (line 196)
Interpretation: Authors frequently fail to clarify the factors that lead to quality changes in the examined lettuce (such as its precise antibacterial action). This has to be completed.
Round 2
Reviewer 1 Report
The manuscript was very well revised by the authors. The authors fully answered all questions of the reviewer. I recommend manuscript for publication.
